# Curcumin Suppresses Lead-Induced Inflammation and Memory Loss in Mouse Model and In Silico Molecular Docking

**DOI:** 10.3390/foods11060856

**Published:** 2022-03-17

**Authors:** Suksan Changlek, Mohammad Nasiruddin Rana, Moe Pwint Phyu, Naymul Karim, Hideyuki J. Majima, Jitbanjong Tangpong

**Affiliations:** 1Biomedical Sciences, School of Allied Health Sciences, Walailak University, Thai Buri 80160, Thailand; suksun.ch@wu.ac.th (S.C.); nasiriiuc09@zju.edu.cn (M.N.R.); moepwintphyu1990@gmail.com (M.P.P.); naiemph@zju.edu.cn (N.K.); k0941761@kadai.jp (H.J.M.); 2Research Excellence Center for Innovation and Health Products (RECIHP), School of Allied Health Sciences, Walailak University, Thai Buri 80160, Thailand; 3Department of Pharmacology, School of Basic Medical Sciences, Zhejiang University, Hangzhou 310058, China; 4College of Biosystems Engineering and Food Science, Zhejiang University, Hangzhou 310058, China

**Keywords:** curcumin, lead, inflammation, neurotoxicity, in silico molecular docking

## Abstract

This study examined the efficacy of curcumin (Cur) against lead (Pb)-induced oxidative damage, inflammation, and cholinergic dysfunction. Institute for Cancer Research (ICR) mice received Pb (II) acetate in drinking water (1%) with or without Cur via oral gavage. Blood and brain tissues were collected for investigation. Pb increased the inflammatory markers and oxidative parameters, which were ameliorated by Cur administration. Cur treatment also improved memory loss, learning deficit, and cholinergic dysfunction via elevating acetylcholinesterase (AChE) enzymatic activity and protein expression. In silico molecular docking supported the results; Cur had a potent binding affinity for AChE receptors, tumor necrosis factor-α (TNF-α), cyclooxygenase-2 (COX-2), phosphorylations of IκB kinase (IKK), extracellular signal-regulated kinase (ERK), c-Jun N-terminal kinase (JNK), and p38 mitogen-activated protein kinase (p38). According to the chemical absorption, distribution, metabolism, excretion, and toxicity (ADMET) profile, Cur could serve as a potential candidate for Pb detoxication substance via exerting antioxidant activity. Taken together, our results suggest that Cur is a natural compound that could be used for the treatment of neurodegenerative disorders via suppressing lead-induced neurotoxicity.

## 1. Introduction

Humans are often exposed to lead (Pb) from water, food contamination, and air pollution caused by industrial emissions. Pb is a notorious xenobiotic that can affect all organs, including the brain [1,2]. In general, the toxic effect of Pb is related to the imbalance between oxidants and antioxidants, termed oxidative stress, which initiates the chain reactions of free radicals through lipid peroxidation [1,3,4]. Evidence suggests that Pb^2^^+^ and Ca^2^^+^ interactions in the mitochondria increase the reactive oxygen species (ROS) level, which further dysregulates the intracellular calcium ([Ca^2^^+^]i) homeostasis and affects Ca^2^^+^-dependent protein mechanisms, thereby causing protein dysfunction [5,6]. Besides this, it impairs the mitochondrial potential to trigger the cellular apoptosis via cytochrome c release and limits antioxidant activities [7,8]. A study by our lab revealed that Pb-induced neurotoxicity causes acetylcholinesterase (AChE) dysfunction related to memory loss and learning deficit [9]. AChE is a serine protease, which is responsible for the hydrolyses of the neurotransmitter acetylcholine (ACh) and is involved in cognitive processes. AChE is mainly available at neuromuscular junctions and cholinergic brain synapses [2,9]. Basha et al. reported that Pb exposure also causes monoamine oxidase (MAO) dysfunction-mediated cognitive impairments [10].

Curcumin (Cur) is a polyphenol primarily extracted from turmeric, and has drawn much attention in the field of natural drug discovery due to its excellent therapeutic effects, such as its antioxidant, anti-inflammation, anti-microbial, anti-arthritic, and anti-depressant activities [11]. In addition, Cur can modulate cognitive dysfunction and astrocyte proliferation [12,13]. Research revealed that Cur can suppress memory impairment [14] and can attenuate cognitive deficits [15]. In mercury chloride-treated offspring mice, Cur treatment improved memory and learning activity, antioxidant profile, and increased AChE, serotonin, and dopamine [16]. Cur also downregulated the AChE gene expressions [17,18]. Apart from Cur attention in the natural drug discovery field, Cur can also be used as an antioxidant supplement, flavoring agent (gives warm and bitter taste), and natural colorant (provides bright yellow color) in different foods or cuisines [19]. Though Cur has potential benefits, several limitations including low water solubility, degradation during processing, etc., reduce the scope of Cur application. Thus, many researchers are trying to increase the functionality of Cur in several foods, drugs, and cosmetic products after successful micro- and/or nano-encapsulation [20].

Interestingly, there is no existing report regarding the in silico molecular docking analysis of Cur against Pb-induced oxidative damage, inflammation, and cognitive dysfunctions in the brain. Therefore, this study aimed to investigate whether Cur could protect against Pb-treated neurotoxicity by alleviating the oxidative stress, inflammation, and AChE dysfunctions.

## 2. Materials and Methods

### 2.1. Chemicals and Reagent

Curcumin (Cur) was supplied by the Research Excellent Center for Innovation and Health Product (RECIHP), Walailak University, Nakhon Si Thammarat, Thailand. All analytical-grade chemicals were bought from Merck KGaA (Darmstadt, Germany) unless otherwise mentioned.

### 2.2. Animals Designing and Treatment

Five groups (*n* = 6) of 30 male ICR mice (31 ± 1 g, 8-week-old) were caged, nurtured, and acclimatized for 7 days in a 12 h:12 h light:dark cycle chamber at 23 ± 2 °C. The mice were handled following the guidelines approved by the Animal Care and Use Committee (ACUC) of Walailak University (No. 002/2013).

In this experiment, lead acetate (Pb, 1% *w*/*v*) was dissolved in drinking water, and sodium acetate (NaAc 1% *w*/*v*) was used to control lead acetate [9,21]. Treatment groups of mice received Cur 100 or 200 mg per kg body weight (BW) daily from 8:30 to 9:30 a.m. through oral gavages. The treatment was conducted for 38 days as follows:
Gr. i: Untreated control   NaAc (1%, *w*/*v*) + normal salineGr. ii: Pb (1%, *w*/*v*)   PbAc (1%, *w*/*v*) + normal salineGr. iii: Pb + Cur 100 mg/kgBW   PbAc (1%, *w*/*v*) + Cur 100 mg/kgBW in normal salineGr. iv: Pb + Cur 200 mg/kgBW   PbAc (1%, *w*/*v*) + Cur 200 mg/kgBW in normal salineGr. v: Cur 200 mg/kgBW   NaAc (1%, *w*/*v*) + Cur 200 mg/kgBW in normal saline


### 2.3. Water Maze Swimming Test

The water maze swimming task (WMST) was performed according to earlier methods [9] to understand memory and learning functions. In this study, each mouse was placed in a 190 × 45 cm circular swimming pool filled with water (25 ± 1 °C). A transparent plastic plinth (6 × 29 cm) was left in one of the pool’s quarters. Training was carried out for 120 s on day 1 in the absence of the plinth. The animals were given two daily trials on the following five days with a 30 min rest between tasks with the submerged plinth. When an animal found the plinth, it was left for 10 s. The mouse was left on the plinth for 10 s if it did not find the plinth within 120 s. The mouse was returned to the cage and patted dry with a paper towel. The time it took to find the submerged plinth in the final trial session, known as the latency time, was recorded.

### 2.4. Forced Swimming Test

A depressive-like behavior was determined by the forced swimming test (FST) following a previously published study [9]. Briefly, the mice were tested in individual 12 × 23 cm vertical glass cylinders. The pool was filled with water (25 ± 1 °C). The immobility times of 2 swimming sessions, including the first 15-min session followed by 24 h later a 6-min test, were recorded.

### 2.5. Blood Sampling and Tissue Harvesting

After the behavioral studies the animals were anesthetized using Nembutal/sodium (65 mg/kgBW), blood was obtained via left ventricle puncture, and perfusion followed using cold phosphate buffer saline at pH 7.4. The blood was added into an ethylenediaminetetraacetic acid (EDTA) tube and centrifuged at 2500× *g* for 15 min to separate plasma and red blood cells for analysis. The brain was removed and homogenized in cold phosphate buffer saline (PBS), containing a mixture of protease inhibitors (leupeptin, pepstatin, and aprotinin) prior to centrifugation at 15,000× *g* for 15 min. The resulting supernatant was collected and kept at −80 °C for further analysis. Brain tissues for immunohistochemistry were rolled on the surface of dry ice wrapped in aluminum foil and quickly put in liquid nitrogen; they were then kept at −80 °C until use.

### 2.6. Measurement of Lipid Peroxidation in Plasma, Red Blood Cell (RBC), and Brain Tissue

Malondialdehyde (MDA) was examined in the samples using a thiobarbituric acid-reactive substances (TBARS) assay [22]. Levels of MDA were compared with a standard 1,1,3,3-tetraethoxypropane. The result was calculated as nM of plasma, µM/g of Hb, and nM/g of protein.

### 2.7. Measurement of Acetylcholinesterase Activity in Plasma, RBC, and Brain Tissue

The acetylcholinesterase (AChE) activity was examined via the approach, with slight modifications, described by Ellman et al. [23]. Briefly, the plasma, RBC, or brain tissue homogenate was mixed with a reaction mixture in phosphate buffer (0.1 M, pH 7.4) containing 5,5-dithiobis-2 nitrobenzoate and acetylthiocholine iodide. The total mixture was then left at 25 °C for 30 min. The degradation of acetylthiocholine iodide was read at 412 nm using a UV–Vis spectrophotometer (Hitachi, Tokyo, Japan). The results were calculated and presented as U/mL for plasma sample, U/g hemoglobin for RBC sample, and U/g protein for brain tissue homogenates following the company protocol.

### 2.8. Enzyme-Linked Immunosorbent Assay (ELISA)

A mouse TNF-α colorimetric sandwich ELISA kit (R&D Systems, Minneapolis, MN, USA) was used to determine the circulating tumor necrosis factor alpha (TNF-α) following the company protocol.

### 2.9. Western Blot Assay

In brief, the brain protein levels were determined by Bradford assay [24], run via sodium dodecyl sulfate (SDS)-polyacrylamide gel electrophoresis (PAGE) at 100 V. The proteins were then transferred to a nitrocellulose membrane at 100 V and 4 °C for 2 h. The membrane was then inhibited for 1 h in Tris-buffered saline-Tween 20 (TBST) buffer, followed by immersion at 4 °C for 24 h with appropriate primary antibodies against AChE (Abcam PLC, Waltham, MA, USA), TNF-α, COX-2, iNOS (Cell Signaling Technology, Danvers, MA, USA) and β-actin (Sigma, Saint Louis, MO, USA). The membrane was then rinsed twice with TBST (7 min) and Tris-phosphate saline (TBS), and then immersed in horseradish peroxidase-conjugated secondary antibody at 25 °C. After rinsing, the protein bands were detected using an enhanced chemiluminescence detection kit (Bio-Rad Laboratories, Inc., Hercules, CA, USA). The results have been expressed as a ratio to the β-actin band using the Kodak GelQuant software.

### 2.10. Immunohistochemistry Study

Brain tissue slices were fixed in 4% paraformaldehyde for 15 min, air-dried, and washed with PBS. Nonspecific proteins were blocked in the blocking serum, consisting of 3% normal donkey serum and 0.3% Triton X-100 in PBS, and incubated at room temperature for 30 min. After blocking, the slices were incubated with primary anti-AChE (Abcam PLC, Waltham, MA, USA). The sections were kept in a humidified box at 4 °C overnight. Tissues were washed three times with PBS and then were incubated for 1 h with donkey antibody-conjugated secondary antibodies conjugated with fluorescent dyes. Excess secondary antibodies were removed by washing three times in PBS and once with deionized H_2_O. Tissue slides were counter-stained with nuclear staining dye (DRAQ5) and mounted with a mounting medium (Vectashield, H-100, Vector Laboratories, Burlingame, CA, USA). Photomicrographs were obtained using a Leica confocal fluorescence microscope (Leica Microsystems Inc., Bannockburn, IL, USA).

### 2.11. In Silico Study

#### 2.11.1. Molecular Docking Study

The required crystal structures were published in the PDB format, as follows: AChE (PDB ID: 4M0E), COX-2 (PDB ID: 5KIR), TNF-α (PDB ID: 2AZ5), IKK (PDB ID: 4KIK), ERK (PDB ID: 1TVO), JNK (PDB ID: 3OY1), and P38 (PDB ID: 5WJJ). These were analyzed using Maestro 11 software (Schrodinger, New York, NY, USA) for the molecular docking study. For protein docking, the protein preparations were allowed to preprocess by adding bond sequence, the H-bond, and SS-bond formation. After that, minimization was achieved using the OPLS5 force field. For ligand processing, the 2D structures of Cur (PubChem CID 969516) were converted into 3D structures using the ligprep software (Scodinger suite v11), with the OPLS5 force field at pH 7.0 ± 2.0 as an initial setting. The prepared proteins were then processed to form a receptor grid using the OPLS5 force field.

#### 2.11.2. The Chemical Absorption, Distribution, Metabolism, Excretion, and Toxicity (ADMET) Analysis

The ADMET analysis and drug likeliness properties of Cur were identified using Schrodinger’s QikProp module. The drug likeliness properties were evaluated based on the rule of five. On the other hand, a candidate was regarded as drug-like if it displayed any three of the parameters’ molecular weight (MW) <500 Daltons, H-bond acceptor (HBA) ≤10, H-bond donor (HBD) ≤5, and predicted QPlogPo/w value (-) 2.0–6.5.

### 2.12. Statistical Analysis

The results were calculated as the mean ± SEM (standard error mean) of three independent experiments. The statistical analysis was conducted using one-way analysis of variance (ANOVA) followed by a post hoc test, Newman–Keuls, using GraphPad Prism v5 (San Diego, CA, USA). *p* < 0.05 was considered statistically significant.

## 3. Results

### 3.1. Effect of Curcumin on Lead-Induced Mice Body Weight Changes

The effects of Cur on animal body weight changes before and after Pb exposure are summarized in Table 1. According to Table 1, at the initial stage, there were no differences between the mice body weights in various treatment groups compared to untreated control. On the contrary, at the final stage, Cur with Pb treatment significantly (*p* < 0.05) improved body weight compared to the lead-treated group. In addition, mice treated with Pb showed less improvement of final body weight compared to Pb-treated initial body weight.

### 3.2. Validation of Curcumin’s Effect against Lead-Induced Cognitive Dysfunction

The Pb-induced cognitive function was evaluated via latency and immobility times. The data are presented in Figure 1. Our WMST revealed no significant differences between Pb-treated mice and others on the first day of the trials (data not shown). In contrast, Pb exposure significantly (*p* < 0.05) increased the latency time (approximately two-fold) on the fifth day of the trial. However, the increase of latency times was reduced significantly by Cur (Figure 1A). Furthermore, the latency time of Cur-treated mice was not significantly different compared to control on the fifth day of trial, which indicates the ameliorative effect of Cur.

The FST study was conducted to evaluate the immobility time of the normal, Pb-, and Cur co-treated mice, and the data are displayed in Figure 1B. Our results showed that co-treatment with Cur significantly (*p* < 0.05) increased the swimming time and reduced the immobility time compared to the lead-treated group. In contrast, Pb-treated mice showed a markedly (*p* < 0.05) prolonged immobility time compared to the control group. Furthermore, there was no significant difference between mice treated with Cur and the control group, indicating an improvement in memory function due to the Cur treatment. These studies confirm that latency and depressive-like behavior are markedly increased in the Pb-treated group.

### 3.3. Protective Effect of Curcumin on Cholinergic Dysfunction

We investigated the acetylcholinesterase (AChE) activity in plasma, RBC, and brain homogenates to unveil Cur’s effect on Pb-treated neurotransmitter impairment. For further clarification, we determined the activity and expression of AChE in the brain tissue. The results regarding cholinesterase function are shown in Figure 2. Figure 2A–C represent the activities of AChE in plasma, RBC, and brain homogenate, respectively. Our study found that the AChE activities of Pb-treated mice were markedly (*p* < 0.05) diminished.

On the contrary, the Pb-treated mice with Cur showed significantly (*p* < 0.05) modulated AChE dysfunction in the plasma, RBC, and brain homogenates (Figure 2A–C). Moreover, the AChE values of Cur co-treatment groups were not significant compared to those of the control group. Afterward, to further confirm the effect of Cur, we determined the expression of AChE in the mouse brain (Figure 3). Figure 3 showed that the levels of AChE expression in the brains of Pb-treated mice were markedly (*p* < 0.05) reduced, while the expressions in Pb-treated mice with Cur were significantly (*p* < 0.05) ameliorated (Figure 3).

The Pb-induced lower AChE expression in brain tissues was confirmed by immunohistochemistry and was ameliorated in our Cur-co-treated group. Additionally, the statistical analysis has shown no significant differences in protein expression between the Cur-treated and the control group, highlighting the potential effect of Cur co-treatment on Pb-intoxicated mice (Figure 4).

### 3.4. Protective Effect of Curcumin on Oxidative Stress Parameters

Malonaldehyde (MDA) is a metabolite of oxidative stress, and is widely accepted as an indicator of lipid peroxidation. Therefore, the MDA levels in plasma, RBC, and brain homogenate were evaluated to understand the effect of Cur against Pb-induced neurotoxicity, and the data are presented in Table 2. As found in Table 2, the MDA in blood and brain of Pb-treated mice was markedly increased compared to the control group, whereas Cur co-treatment reduced the MDA in a concentration-dependent manner.

### 3.5. Protective Effect of Curcumin on Inflammatory Profile

The plasma TNF-α level, an expression of inflammatory markers in the brain homogenate, was estimated to confirm the anti-inflammatory effect of Cur against Pb-induced neurotoxicity. The results are shown in Figure 5. According to Figure 5A, the circulating TNF-α level was markedly (*p* < 0.05) increased in Pb-treated mice. On the flip side, Cur at a dose of 100 or 200 mg/kgBW to Pb-treated mice significantly (*p* < 0.05) reduced the plasma TNF-α levels (Figure 5A). Similarly, the expressions of inflammatory markers (TNF-α, COX-2, and iNOS) were higher in the brain tissue homogenate of Pb-treated brain (Figure 5B–D). Additionally, co-treatment with Cur significantly (*p* < 0.05) reduced the expressions of inflammatory markers (Figure 5B–D).

### 3.6. Molecular Docking In Silico

#### 3.6.1. Binding Mechanism of Curcumin with Acetylcholinesterase (AChE)

In silico molecular docking showed Cur’s potent binding affinity for AChE receptors (PDB ID: 4M0E), COX-2 (PDB ID: 5KIR), TNF-α (PDB ID: 2AZ5), IKK (PDB ID: 4KIK), ERK (PDB ID: 1TVO), JNK (PDB ID: 3OY1), and P38 (PDB ID: 5WJJ) (Table 3 and Figure 6). According to Table 3, Cur showed a greater ability to dock with AChE, followed by IKK, COX-2, P38, JNK, TNF-α, and ERK. Thus, Cur was predicted to have neuroprotective effects by exerting its antioxidant and anti-inflammatory properties, and by inhibiting cholinergic alterations.

#### 3.6.2. ADMET Analysis and Drug-Likeliness Properties of Curcumin

According to the ADMET analysis (Table 4), Cur could serve as a potential candidate, along with existing drugs. The ADMET profile of Cur meets the necessary criteria to become a drug candidate: a lower star (*:1), with moderate lipid solubility comprising characteristics such as FOSA (261.70), QPlogBB (−2.246), and QPPMDCK (65.952). Furthermore, to be an orally available drug candidate, the substance must follow Jorgensen’s rule of three and Lipinski’s rule of five. In addition, our QikProp module analysis suggests that Cur also has a modest ability to cross the gut barrier (QPPCaco 155.011) following oral intake. Therefore, Cur could be regarded as a potential candidate for further analysis in the clinical environment.

Here, a low star value indicates that a molecule is more drug-like than other molecules with many stars. The total solvent accessible surface area (SASA) in Å^2^ was measured using a probe with a 1.4 Å radius, the FOSA-hydrophobic component of the SASA (saturated C and attached H), and FISA-hydrophilic part of the SASA (SASA on N and O, and H on heteroatoms).

Here, Leu: leucine, Trp: tryptophan, Ala: alanine, Val: valine, Ile: isoleucine, Met: methionine, Cys: cysteine, Pro: proline, Tyr: tyrosine, Phe: phenylalanine, Lys: lysine, Arg: arginine, Thr: threonine, Gln: glutamine, Ser: serine, Asn: asparagine, Gly: glycine, His: crystal histidine. In addition, there are three residue templates for histidine such as Hie: histidine (default, H on the delta nitrogen), HID (H on the epsilon nitrogen), HIP (hydrogens on both nitrogens, this is positively charged).

## 4. Discussion

Lead (Pb) is a ubiquitous metal that causes decreases in body weight and has been implicated in many other organs; in particular, the nervous system is the primary target of Pb exposure, and the developing brain appears to be especially vulnerable to Pb neurotoxicity [25,26]. It has been observed that Pb exposure is associated with a decline in attention concentration, the inhibition of long-term potentiation (LTP), growth retardation, and learning and memory impairment [27,28]. Water maze swimming tests are related to spatial memory and are a popular method used to evaluate cognitive function in animal models, including Alzheimer’ disease (AD) model mice [29,30]. Acetylcholine exclusively encodes the regulation of spatial memory through theta rhythm oscillations. Besides this, acetylcholine processes and encodes novel information for short-term and long-term memory through persistent spiking (such as cortical neurons) [31,32]. Our current results show that Cur improved behavioral changes, including learning deficits, memory losses, and stress in mice. In addition, Cur has been reported to be able to repair learning and memory dysfunctions [33]. Similar studies have also confirmed Cur’s effect against learning and memory impairments [34,35].

The present study reports that the structure of AChE contains free SH groups, which have a high affinity for Pb. After binding Pb with the free SH groups of AChE, it can alter Ca^2^^+^ homeostasis and produce oxidative stress [2,36,37]. The reductions in AChE expression in Pb-treated brains were reversed via co-treatment with Cur, indicating improvements in cholinergic circuitry. At the cholinergic synapses, AChE hydrolyzes the neurotransmitter acetylcholine. Dysfunctions in AChE cause neuropsychiatric problems [17,34]. The over-activation of the cholinergic system hinders working memory activity in the primate prefrontal cortex [38]. Hence, the hippocampal–prefrontal axis assists in spatial encoding within the working memory, possibly via gamma oscillations synchrony [39]. On the contrary, reduced acetylcholine results in cognitive impairments. Thus, there should be a balance, as acetylcholine may be friendly or antagonistic in different contexts, as Baxter and Crimins discussed [40]. Therefore, the current study showed that Cur markedly restored AChE expression, which might help in establishing a balance in regular acetylcholine accumulation in Pb-intoxicated mice.

Studies have reported that Pb causes oxidative damage by generating free radicals, reducing the cellular antioxidant pool, and manipulating several pathways. It further diminishes the glutathione and protein-bound sulfhydryl group [1,9]. Cur is known to have a potent antioxidant activity due to its capacity for chain-breaking, as well as the hydrogen-donating phenolic groups in its structure [41]. Much research has also revealed the neuroprotective effects of Cur, which operate via exerting antioxidant effects and reducing oxidative damage [18,34,37,41,42]. Besides this, Pb accelerates caspase-3 activity via oxidative stress, lipid peroxidation, and apoptosis [7,43]. Curcumin exhibits great promise as a therapeutic agent for a variety of cancers, as well as for psoriasis, and Alzheimer’s disease [44]. It has been reported that Cur suppresses aluminum- and Pb-induced oxidative neurotoxicity, alterations in NMDA receptors that lead to decreased antioxidant enzyme activity, and AChE dysfunctions [45]. These two possible mechanisms (oxidative stress and cholinergic dysfunction) relate to Pb-induced neurotoxicity. Thus, the present study has indicated that Cur, a potent antioxidant compound, ameliorated lipid peroxidation and oxidative stress parameters in mice exposed to Pb, along with protecting against cholinergic dysfunction.

Accumulating evidence is suggesting that inflammation is the main outcome of Pb contamination. Pb intoxication causes ROS-mediated inflammation by activating the JNK–MAPK pathway. Furthermore, ROS-activated NF–κB is translocated to a nucleus in order to initiate inflammatory gene transcription [21,41,46]. Thus, Pb could induce neuroinflammation via regulating the NF–κB pathway, thereby increasing inflammatory gene expression and enzyme alterations [41,47,48,49], while Cur co-treatment effectively suppresses Pb-induced brain inflammation.

In silico molecular docking showed that Cur had a greater ability to dock with AchE, followed by IKK, COX-2, P38, JNK, TNF-α, and ERK. Thus, Cur may completely bind with AchE to increase enzymatic activity, which is predicted to have dose-dependent neuroprotective effects, at least in part via exerting antioxidant and anti-inflammatory effects, and reversing the Pb-induced alterations in transmitters and enzymes.

## 5. Conclusions

Our findings suggest that Pb intoxication causes memory and learning deficits associated with aberrant acetylcholine balance. However, co-treatment with Cur restored Pb-induced alterations in cognitive processes by maintaining the cholinergic system via rebalancing the acetylcholine level. Therefore, following our thorough investigation, the dietary intake of Cur could be suggested to prevent and cure heavy metal-associated memory loss, learning deficit, and depression-like behavior.

## Figures and Tables

**Figure 1 foods-11-00856-f001:**
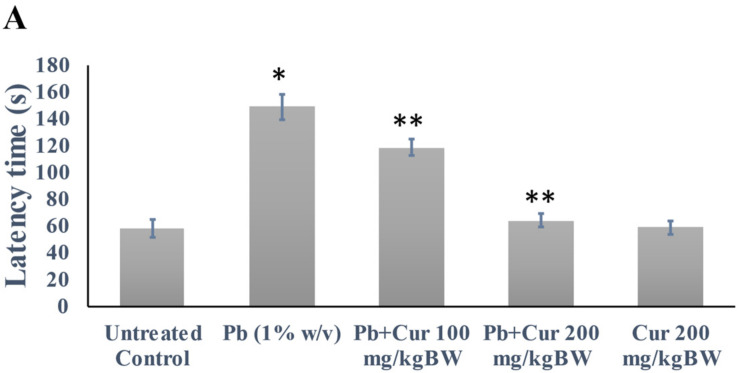
Effect of curcumin on cognitive dysfunction caused by Pb. (**A**) Curcumin’s effect against memory loss and learning deficit of mice as related to latency time (s) using the Morris water maze swimming test. (**B**) Curcumin’s effect against depression-like behavior of mice as related to the immobility time (s) using the forced swimming Test. The results are displayed as mean ± SEM (*n* = 6). * *p* < 0.05 specifies a difference between control mice and Pb-treated mice. ** *p* < 0.05 compares Pb-treated mice and curcumin (Cur)-treated mice.

**Figure 2 foods-11-00856-f002:**
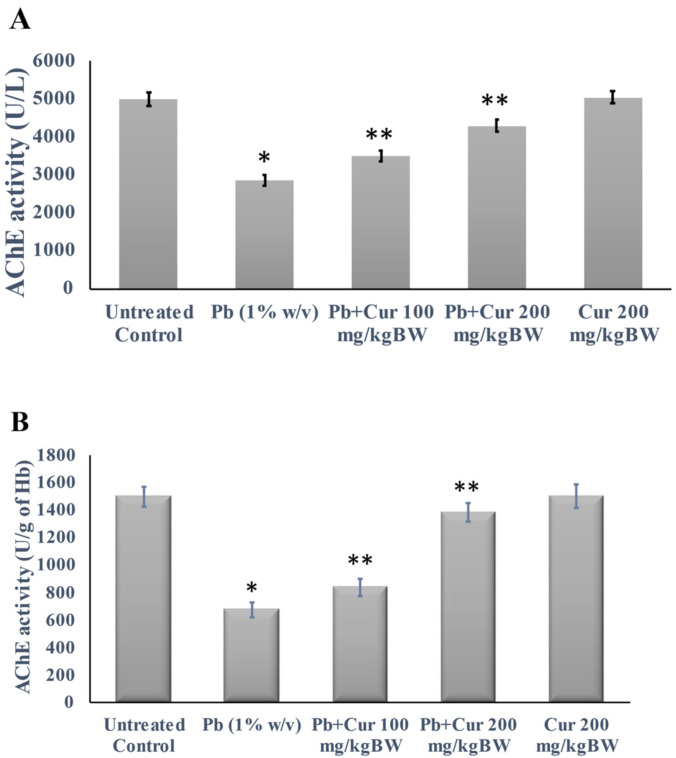
Effect of curcumin on cholinergic dysfunction caused by lead. (**A**–**C**) Acetylcholinesterase (AChE) activity in the plasma, RBC, and brain homogenate, respectively. The results are displayed as mean ± SEM (*n* = 6). * *p* < 0.05 specifies a difference between control mice and Pb-treated mice. ** *p* < 0.05 compares between Pb-treated mice and Curcumin (Cur)-treated mice.

**Figure 3 foods-11-00856-f003:**
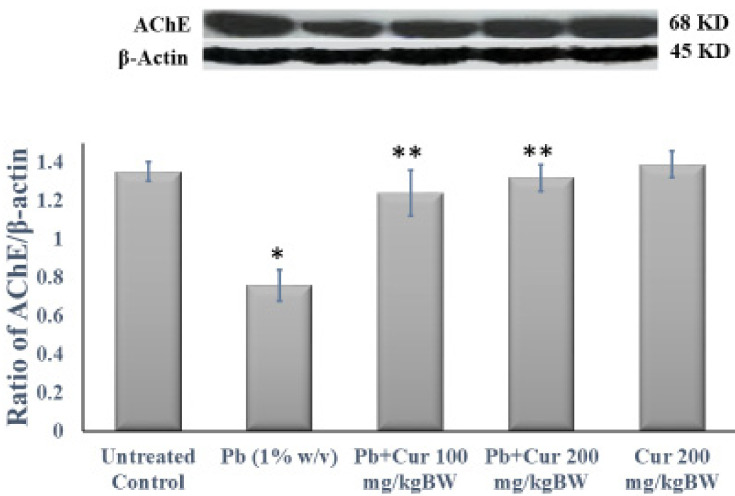
Effect of curcumin on the expression of acetylcholinesterase (AChE) in Pb-treated mouse brain. The internal control of protein bands was examined using β-actin in a relative density analysis. The results are displayed as mean ± SEM (*n* = 6). * *p* < 0.05 specifies difference between control mice and Pb-treated mice. ** *p* < 0.05 compares between Pb-treated mice and curcumin (Cur)-treated mice.

**Figure 4 foods-11-00856-f004:**
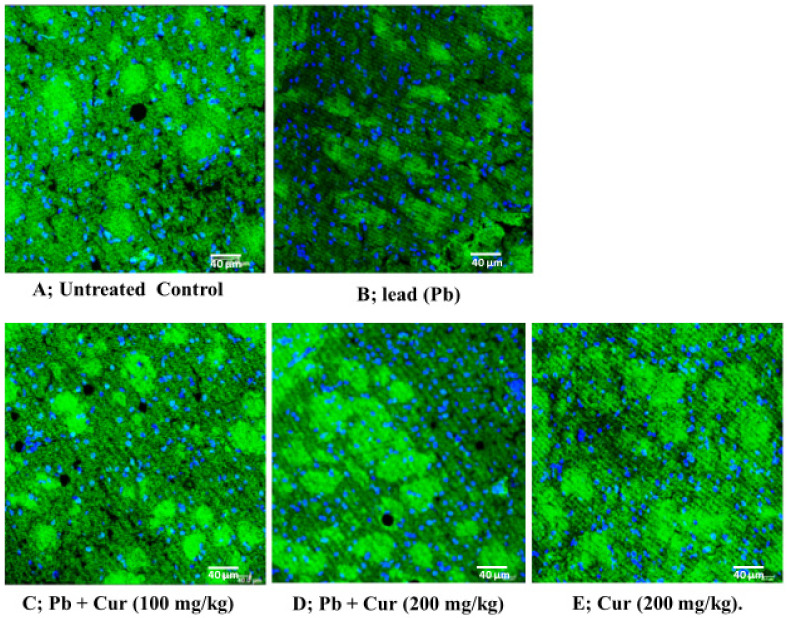
Immunohistochemistry—effect of curcumin on the acetylcholinesterase (AChE) expression in the brain tissue of lead (Pb)-exposed mice. (**A**) Group 1: Untreated control (sodium acetate, CH_3_COONa-3H_2_O) in drinking water. (**B**) Group 2: Pb (1%) in drinking water. (**C**) Group 3: Pb + Cur (100 mg/kg). (**D**) Group 4: Pb + Cur (200 mg/kg). (**E**) Group 5: Cur (200 mg/kg). In this figure, the green color represents AChE expression and blue color shows nucleus staining with DRAQ5.

**Figure 5 foods-11-00856-f005:**
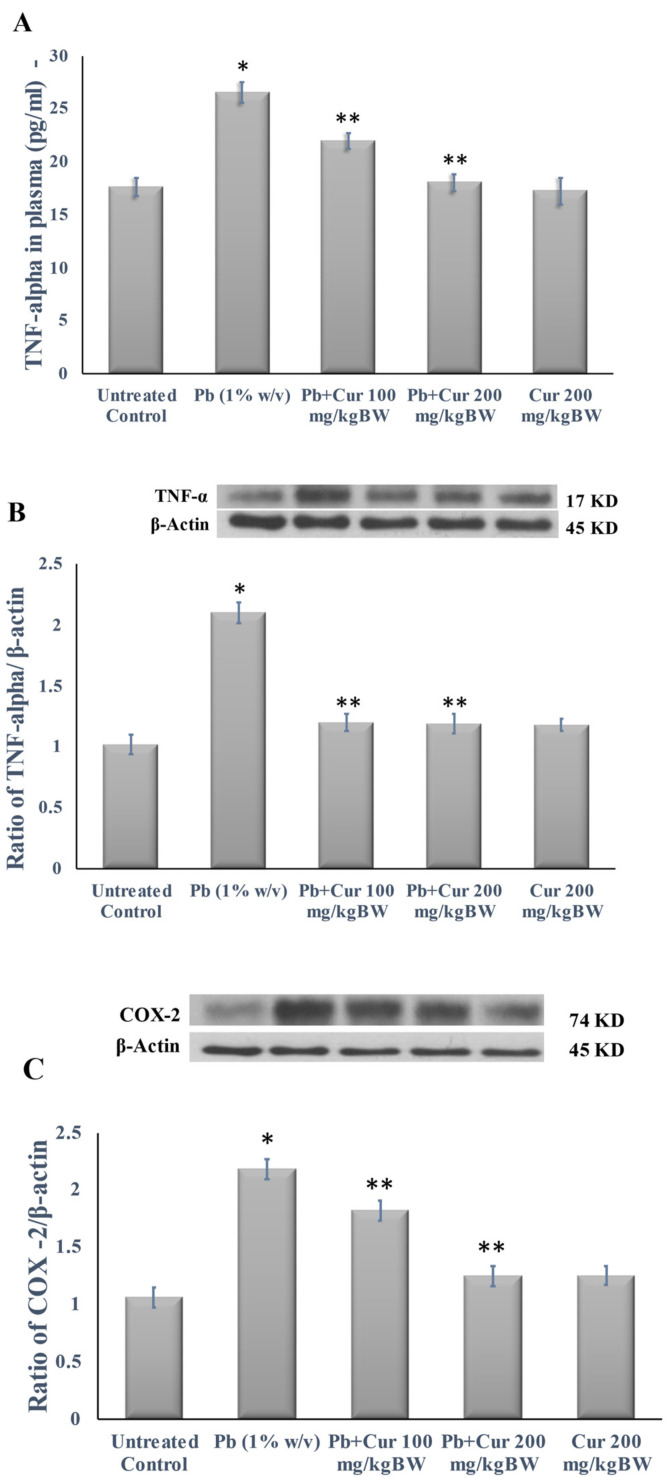
Effect of curcumin on inflammation caused by lead. (**A**) TNF-α level in plasma. (**B**–**D**) expressions of TNF-α, COX-2, and iNOS in brain homogenate, respectively. The results are displayed as mean ± SEM (*n* = 6). * *p* < 0.05 specifies the differences between control mice and Pb-treated mice. ** *p* < 0.05 compares between Pb-treated mice and curcumin (Cur)-treated mice.

**Figure 6 foods-11-00856-f006:**
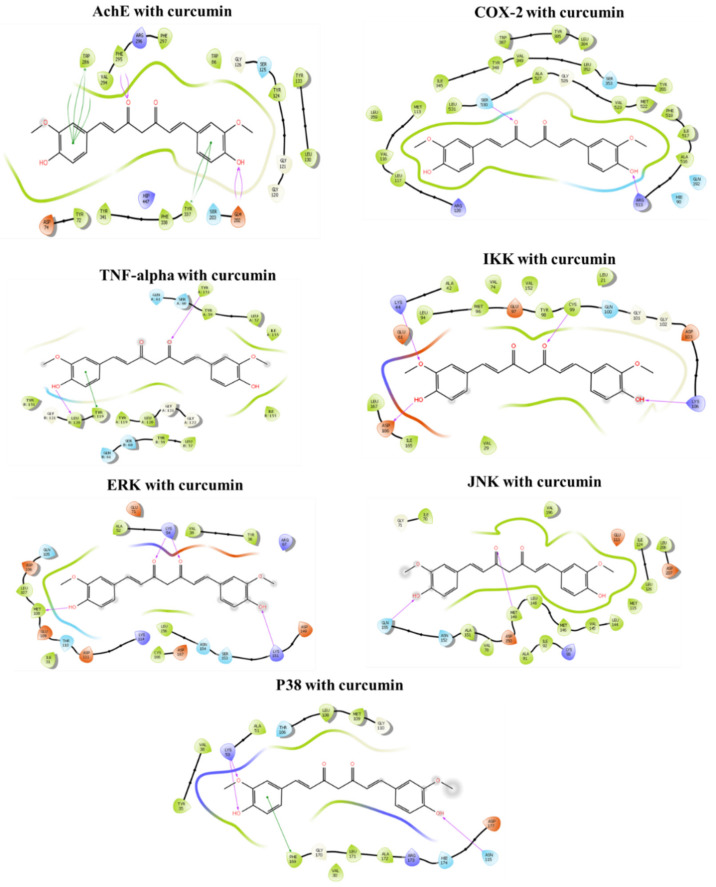
Molecular docking of curcumin with AChE (PDB ID: 4M0E), COX-2 (PDB ID: 5KIR), TNF-α (PDB ID: 2AZ5), IKK (PDB ID: 4KIK), ERK (PDB ID: 1TVO), JNK (PDB ID: 3OY1), and P38 (PDB ID: 5WJJ). Here, the colors indicate the residue: orange—acidic (Asp, Glu); green—hydrophobic (Leu, Trp, Ala, Val, Ile, Met, Cys, Pro, Tyr, Phe); purple—basic (Lys, His, Arg), blue—polar (Thr, Gln, Ser, Asn, Hie, His, Hid); grey (Gly and water). The line between receptor and ligand indicates the interaction: pink (H-bonds to the protein backbone); orange (π-cation interaction); green (π-π stacking interactions). The protein “pocket” is visualized with the colored line around the ligand.

**Table 1 foods-11-00856-t001:** Effect of curcumin on mice body weight changes in lead-exposed mice.

Group	Initial Body Weight (g)	Final Body Weight (g)
Untreated control	29.53 ± 0.35	37.16 ± 1.01
Pb (1%, *w*/*v*)	30.01 ± 0.37	32.80 ± 0.58 *
Pb + Cur 100 mg/kgBW	30.78 ± 0.37	36.50 ± 0.71 **
Pb + Cur 200 mg/kgBW	30.25 ± 0.46	38.69 ± 0.58 **
Cur 200 mg/kgBW	30.22 ± 0.38	37.50 ± 0.80 **

The results are shown as mean ± SEM (*n* = 6). * *p* < 0.05 specifies the differences between control mice and Pb-treated mice. ** *p* < 0.05 compared to Pb-treated mice vs. curcumin (Cur)-treated mice.

**Table 2 foods-11-00856-t002:** The effects of curcumin treatment against lead-induced oxidative stress on malondialdehyde levels in plasma, RBC, and brain tissues after treatment 38 days.

Group	Plasma MDA (nM)	RBC MDA (µM/g Hb)	Brain MDA (nM/g Protein)
Untreated control	91.43 ± 3.42	3.72 ± 0.18	37.67 ± 3.43
Pb (1%, *w*/*v*)	178.74 ± 4.54 *	6.98 ± 0.44 *	50.80 ± 2.71 *
Pb + Cur 100 mg/kgBW	125.78 ± 3.21 **	4.14 ± 0.94 **	42.08 ± 0.62 **
Pb + Cur 200 mg/kgBW	103.94 ± 5.76 **	3.65 ± 0.20 **	39.02 ± 1.93 **
Cur 200 mg/kgBW	87.83 ± 2.12 **	3.89 ± 0.11 **	38.88 ± 1.46 **

The results are shown as mean ± SEM (*n* = 6). * *p* < 0.05 specifies a difference between control mice and Pb-treated mice. ** *p* < 0.05 compares between Pb-treated mice and curcumin (Cur)-treated mice.

**Table 3 foods-11-00856-t003:** Curcumin attenuates oxidative stress, inflammation, and acetylcholinesterase dysfunction via interacting with AChE, COX-2, TNF-α, IKK, ERK, p38, and JNK.

Compound	Protein	PDB ID	Docking Score	Glide e-Model	Glide Energy
Curcumin	AChE	4M0E	−9.251	−75.443	−52.127
	COX-2	5KIR	−8.113	−55.858	−42.837
	TNF-α	2AZ5	−6.857	−59.754	−43.063
	IKK	4KIK	−8.837	−85.148	−56.14
	ERK	1TVO	−5.479	−62.176	−46.786
	JNK	3OY1	−7.172	−68.542	−50.149
	P38	5WJJ	−7.739	−70.85	−49.057

AChE; acetylcholinesterase, COX-2; cyclooxygenase-2 (COX-2), TNF-α; tumor necrosis factor-α (TNF-α), phosphorylations of IκB kinase (IKK), extracellular signal-regulated kinase (ERK), p38 mito-gen-activated protein kinase (p38), and c-Jun N-terminal kinase (JNK).

**Table 4 foods-11-00856-t004:** ADMET analysis of Curcumin.

Compound	Curcumin
Star	1
SASA	706.80
FOSA	261.70
FISA	190.40
QPPCaco	155.01
QPlogBB	−2.246
QPPMDCK	65.96
Rule of three	0
Rule of five	0

SASA; total solvent accessible surface area, FOSA, hydrophobic SASA, FISA; hydrophilic SASA, QPPCaco; Caco-2 cell permeability, QPlogBB; predicted brain/blood partition coefficient, QPPMDCK; predicted apparent MDCK cell permeability.

## Data Availability

The data presented in this study are available on request from the corresponding author.

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
