# Peer review of "Curcumin Suppresses Lead-Induced Inflammation and Memory Loss in Mouse Model and In Silico Molecular Docking"

_foods, 2022, doi:10.3390/foods11060856_

Round 1

Reviewer 1 Report

In this study, the authors investigated whether curcumin could protect against neurotoxicity by alleviating the oxidative stress, inflammation, and AChE dysfunctions caused by Pb in ICR mice. The paper is well and very clearly written, and I have only several suggestions:

In introduction:  Please, explain in one sentence the role of acetylcholinesterase.

Line 52. Does curcumin increase the activity of AChE?

Line 113. Please, add the name of the authors before reference 21.

Line 176. Please, add which groups compared in this sentence: However, mice treated with Pb showed significantly (P<0.05) lower body weights.

Line 177. Please, add which groups you compared: On the contrary, Cur with Pb significantly improved body weight compared to the lead-treated group.

Line 187. datanot shown, please separate data and not

Author Response

We would like to respond to reviewer's comments and suggestions as attached file

Reviewer 2 Report

The manuscript shows interesting results. The authors show that curcumin could reduce the toxicity of lead. However, the authors could improve the manuscript by incorporating the potential application of the results in the field of food science and technology. The discussion focuses only on the biochemical aspects of the results and this reduces their impact and is thus not entirely suitable for the review "foods".

Line 16. Define the abbreviation ICR in the abstract section and do the same for all abbreviations presented in the manuscript.

Introduction

Line 34. Please revise the reference number, reference number 1 does not appear in the first citation.

The state of the art presented focuses on the mechanisms of action of Pb toxicity and Cur benefits, but there is a need to further explore the use of the potential application of Cur in food, not only in its biochemical function but also in product development or technological application.

Materials and methods

Line 66. Please adjust the population size used. Five groups (N=5 and not 6) of 42 male ICR mice (31±1g, 8 weeks) were caged, fed…

Line 72. Was the Cur used a commercial product? or is it a turmeric extract? Define BW, are the bases wet?

Line 101. Please define the abbreviation PBS.

Line 111. Measurement of acetylcholinesterase activity in plasma, red blood cells and brain tissue.

A standard curve was performed to determine the plasma U/mL of AChE activity ?

Line 168. Statistical analysis. If you have a control treatment, why don't you use Dunnett's test as a post hoc test?Results

Line 175-16. "According to Table 1, there was no difference between the initial body weights of the mice in the different groups." Is it possible to improve the presentation of the results in the form of tables and figures? It is confusing to have two comparison groups of means.

The electrophoresis image presented could be improved.

Figure 6. The amino acid names in the figure are difficult to follow

.Discussion

As in the introductory section, the discussion focused on biochemical details. However, there is a lack of discussion on the potential application of the results in the dietary field, the scope of this journal. How could Cur be used in food? as a dietary supplement? as a food ingredient? Are Cur's bioactive compounds stable under food processing conditions ?

Author Response

We would like to respond to reviewer's comments and suggestions as attached file.
